# A study protocol for predictors of post-discharge mortality among children aged 5–14 years admitted to tertiary hospitals in Tanzania: A prospective observational cohort study

**Elton Roman Meleki**[1]*, **Stella Mongella**[2]*, **Francis Fredrick Furia**[1]*

**1** Department of Pediatrics and Child Health Muhimbili University of Health and Allied Sciences, Dar-es-Salaam, Tanzania, **2** Department of Pediatrics and Child Health Jakaya Kikwete Cardiac Institute, Dar-es-Salaam, Tanzania

* ellierolland@gmail.com (ERM); mongellasms@gmail.com (SM); Fredrick.francis78@gmail.com (FFF)

## Abstract

### Introduction

Globally, millions of children and adolescents die every year from treatable and preventable causes. Sub-Saharan Africa accounted for 55% of deaths of children aged 5–14 years in 2017. Despite this high burden, minimal effort has been directed toward reducing mortality among older children and adolescents in comparison to under-fives. Mortality rates of children post-discharge vary between 1–18% in limited-resource countries and are reported to exceed in-hospital mortality. In Tanzania, there is limited data regarding post-discharge mortality and its predictors among children aged 5–14 years.

### Objectives

This study aims to determine the post-discharge mortality rate and its predictors among children aged 5–14 years admitted to pediatric wards at MNH, MOI, and JKCI.

### Methods and analysis

This will be a prospective observational cohort study that will be conducted among children aged 5–14 years admitted to pediatric wards at Muhimbili National Hospital, Jakaya Kikwete Cardiac Institue, and Muhimbili Orthopedic Institue in Dar-Es-Salaam, Tanzania. Data will be collected using a structured questionnaire and will include socio-demographic characteristics, clinical factors, and patients' outcomes. Post-discharge follow-up will be done at months 1, 2, and 3 after discharge via phone call. Data will be analyzed using SPSS version 23. The association of demographic, social economic, and clinical factors with the outcome of all causes, 3 months post-discharge mortality will be determined by Cox regression, and survival rates will be displayed through Kaplan-Meier curves.

**Data Availability Statement:** No datasets were generated or analysed during the current study. All

relevant data from this study will be made available upon study completion.

**Funding:** The authors received no specific funding for this work.

**Competing interests:** The authors have declared that no competing interests exist.

**Abbreviations:** GCS, Glasgow Coma Scale; HB, Hemoglobin; HINARI, Health Inter-Network Access to Research Initiatives; HIV, - Human Immunodeficiency Virus; JKCI, Jakaya Kikwete Cardiac Institute; MNH, Muhimbili National Hospital; MOI, Muhimbili Orthopedic Institute; PubMed, Public/Publisher Medline; SPSS, Statistical Package for Social Sciences; WHO, World Health Organization.

## Discussion

This study will determine post-discharge mortality among children aged 5–14 years and its predictors in Tanzania. This information is expected to provide baseline data that will be useful for raising awareness of clinicians on how to prioritize and plan a proper follow-up of children following hospital discharge. These data may also be used to guide policy development to address and reduce the high burden of older children and adolescent mortality and may be used for future studies including those aiming to develop prediction models for post-discharge mortality among older children and adolescents.

## Background

Over the last thirty years, there has been a significant decline in mortality among children <5 years with nearly all global regions implementing Sustainable Development Goals (SDG) reducing mortality among children <5 years by almost half. The same has also occurred for older adolescents and youths aged 15–24 years whose trend of mortality has declined significantly similar to that of under-five [1–3]. On the other hand, mortality reduction in children between 5–14 years has remained stagnant for the same duration of time and no significant interventions have been put forward [1, 4, 5].

Approximately 1 million children aged 5–14 years died in 2017 with nearly all these deaths occurring in lower- and middle-income countries with more than half involving children in Sub-Saharan Africa [2, 6]. Causes of mortality among older children and young adolescents differ across the globe, most deaths are attributed to injuries, malignancy, congenital defects, and infectious causes including enteric infections, neglected tropical diseases, meningitis, and malaria [1, 5, 7–9]. Infectious causes are predominant in lower- and middle-income countries, particularly in Sub-Saharan Africa [5, 10–13].

Improvement of health services has supported early detection and treatment of most clinical conditions during hospitalization with improved survival and discharge [12, 14]. There is limited information on the mortality after discharge, especially for children aged 5–14 years. Several studies have documented post-discharge mortality among children in Sub-Saharan Africa, however, there is a lack of disaggregated data for children aged 5–14 years. The limited available data have provided useful information for children below 5 years including clinical profile, socio-demographic features and major predicting factors for mortality [12, 15–18].

This study will determine post-discharge mortality and its predicting factors among children aged 5–14 years in Tanzania. This study will provide important information to inform practicing clinicians, policymakers, and other interested stakeholders involved in the provision of care for children in Tanzania and the Sub-Saharan Africa region. This information will also provide baseline data for future studies including those aiming at developing a prediction model for post-discharge mortality for older children and adolescents in the Sub-Saharan Africa region.

## Research questions

1. What is the mortality for children aged 5–14 years within 3 months following discharge from pediatric wards at MNH, JKCI, and MOI?

2. What are the predictors of post-discharge mortality among children aged 5–14 years admitted to pediatric wards at MNH, JKCI, and MOI?

## Methodology/Design

### Study design and aim

A prospective observational cohort study will be conducted in pediatric departments of Muhimbili National Hospital (MNH), Jakaya Kikwete Cardiac Institute (JKCI), and Muhimbili Orthopedics Institute (MOI) to analyze mortality at 3 months post-discharge and its predictors among children aged 5–14 years.

### Study setting

This study will be conducted at MNH, JKCI, and MOI which are public National institutes of health housed within the same grounds in Dar-Es-Salaam. They all serve as teaching hospitals for Muhimbili University of Health and Allied Sciences (MUHAS). JKCI and MOI are superspecialized hospitals providing cardiac, neurosurgical, and trauma tertiary care services respectively. The pediatric units in all three hospitals admit children from 0 days to 14 years. The three institutes were chosen because they receive patients from all over Tanzania and the results can be generalized. They also represent different specialties to reflect children of all diagnoses.

MNH has a bed capacity of more than 1500 and attends to about 2,000 outpatients per day. The pediatric unit at MNH has 6 units namely; general pediatrics, acute care unit, infectious/diarrhea unit, malnutrition unit, hematology-oncology unit, and neonatal unit. The pediatric building also houses other specialized units such as surgery, ophthalmology, burn, dental, ENT, and urology for children.

JKCI has a 103-bed capacity and 198 staff attending on average 700 outpatients and 100 inpatients per week while MOI has a bed capacity of 362, with over 100 medical specialists attending about 1000 outpatients per week. The building of MOI houses two specialized pediatric units which are the trauma, the orthopedic unit, and the neurosurgical unit.

### Study population and sample size

This study will enroll all children aged 5–14 years before discharge from pediatric wards at the three hospitals after they provide assent and their caregivers' consent to participation. The sample size was calculated from open Epi using the Fleiss method for rates and proportions [19]. The candidate variables for clinical prediction were considered in sample size calculation utilizing findings from a previous study conducted in Northern Tanzania [15].

The variables included were low hemoglobin level, proteinuria, hematuria, having a diagnosis of cancer, heart disease, sickle cell disease, neurological disease, and clinical findings of higher respiratory rate, low oxygen saturation, low GCS, reduced urine output, and lower limb edema. All the variables were entered into the formula comparing outcomes among exposed and unexposed. The variable that gave the largest sample size was selected. The post-discharge mortality from those who presented with hematuria which was 40% gave the largest sample size, and hence it was selected.

A power of 80% was selected with a two-sided confidence level of 95%. The ratio of unexposed to exposed was 24.3. The percentage of those exposed with an outcome that in the selected study was the mortality among those who had hematuria on urinalysis was 40%. The percentage of unexposed with outcome was 16%. The odds ratio was 2.5. The sample size of exposed was 28 and of unexposed 669 making a total of 697. The sample size was adjusted for a 5% non-response rate and the final sample size was 734 participants. The participants will be recruited conveniently before their discharge.

**Table 1. Candidate variables for predicting post-discharge mortality.**

| Demographics | Social economic factors | Admission type | Clinical data |
|---|---|---|---|
| • Age of the child<br>• Sex of the child<br>• Caregiver's level of education<br>• Caregiver's age<br>• Caregiver's marital status | • Caregiver's employment status<br>• Type of toilet used in the home<br>• Type of material the floor, wall and roof of the house is made of. | • self-referral<br>• Admission from clinic<br>• Referral from another hospital | • Admitting complains<br>• Vitals at admission<br>• Duration of hospital stay<br>• Duration of illness<br>• Prior treatment<br>• Clinical signs<br>• Laboratory parameters<br>• Abnormal imaging<br>• Type of discharge (absconded, against medical advice, or medical) |

## Variables

The variables have been selected based on the literature. Table 1 below represents predicting variables. There is evidence that demographic characteristics are predictors of mortality. A recent systematic analysis of the global burden of diseases revealed age and sex variation in terms of mortality in children. The age between 10–14 years is considered early adolescence and children at this age are more prone to make poor decisions in highly emotionally charged situations compared to children aged 5–9 years [20].

Similarly, educational status of the caregivers has also been shown to predict mortality, and this was reported from a study done in the Philippines where children born to less educated mothers had higher risks of post-discharge mortality. The same was reported in the study that was done in Guinea [18].

There is also evidence of socioeconomic status and clinical characteristics as predictors of mortality from various studies. A few available studies have reported poor housing quality, length of hospital stay, discharge against medical advice, presence of chronic illnesses, and severe anemia amongst others as predictors of mortality [12, 16, 17, 21].

Hence, there is clear evidence that socioeconomic, demographics, and clinical characteristics can predict mortality. However, a majority of this literature does not have disaggregated data for children aged 5–14 years. The variables selected in Table 1 below are going to be evaluated to determine the predictors of mortality in this age group of children.

The outcome variables presented on Table 2 below have also been selected based on literature. The global programs are all aimed at reducing childhood and youth mortality. There is compelling evidence that post-discharge mortality exceeds in-hospital mortality in Sub-Saharan Africa however there is no disaggregated data for children aged 5–14 years [22]. The available data provides useful information for children below the age of five. The primary outcome as shown on Table 2 below is post-discharge mortality, but we also assess other outcomes such as readmission, unplanned hospital visits and use of herbal medications.

## Data collection

Data will be collected by three research assistants and the study's primary investigator. The three assistants will be medical officers who are not directly involved in the care of the patient.

**Table 2. Outcome variables.**

| Primary outcome | Secondary outcomes |
|---|---|
| post-discharge mortality at 3 months | • Readmission<br>• Unplanned hospital visits<br>• Use of herbal medication |

The research assistants will be familiarized with the questionnaire by going through the questions together with the study's primary investigator. A pilot will be conducted to evaluate the questionnaire and competencies of the research assistants. Study participants will be identified prior the discharge and those eligible will be enrolled.

A structured questionnaire will be used for data collection, which will include the demographics of both the caregiver and the participant, socio-economic factors of the caregiver, participants' in-patient clinical information at admission and during the hospital stay, and outpatient outcomes. The questionnaire will also include the caregiver's contacts. The patient's clinical information in the questionnaire will be extracted by the research assistants from the patient's file, while demographics, socioeconomic status and post-discharge follow-up will require interviewing the participant. The interviews will be done by the research assistants and the information will be recorded in the questionnaire.

The in-patient clinical data will include presenting signs and symptoms, and laboratory and imaging results which all shall be recorded in the questionnaire from the patient's progress notes, laboratory, and radiology electronic systems respectively. The outpatient follow-up will be done at months 1, 2, and 3 after discharge from the hospital.

Follow-up will be through a phone call through which a questionnaire will be administered through the phone. The research assistants will be calling the study participants at months 1, 2, and 3 after discharge. The follow-up questionnaire will include the status of the participant if alive or deceased, the general condition of the participant, readmission, and unplanned hospital/outpatient visits. For the deceased individuals, a modified verbal autopsy will be conducted over the phone to establish the possible cause of death by the study's primary investigator.

This modified verbal autopsy will consist of questions that aim at establishing the possible cause of death of the deceased. This will be administered over the phone to the caregiver/parent who was living with the child at the time of death.

## Data collection tools

Data will be collected using structured questionnaires, which will be administered by the research assistants. The questionnaire will be administered through interviews and some other information will be recorded from the patient's progress notes, laboratory and radiological results. The questionnaires are designed based on the research objectives to capture the most complete and accurate information possible. The questionnaires have also been designed to make it easy for the respondent to provide the necessary information and for the interviewer to record an accurate response. Data on the causes of death will be collected using a modified WHO verbal autopsy tool which is designed to capture the age of the participant at death, the place of death, general signs and symptoms of the illness that preceded death, and any treatment given before death [23]. It will also capture the possible causes that lead to the death of the deceased.

## Data management

All data that is collected on paper-based forms will be stored by the primary investigator in a locked private cupboard that only research personnel will be able to access under the primary investigator's permission. Research assistants together with the primary investigator will transfer the information from the paper forms to feed it to the SPSS software version 23. The files will later be merged for data cleaning and analysis.

Information will be confidential and no identifying information will be published or disseminated during the study period or upon completion of the study. Data will only be stored until publication, if the need arises for verification and paper forms will be burnt after the analysis is complete and the report has been written.

## Data analysis

Data will be entered into SPSS version 23. Categorical variables will be described as proportions (percentage), and continuous variables as means with standard deviations. Univariable and multivariable Cox regression analysis will be used to determine factors associated with mortality. All factors that will be significantly associated with mortality in the univariable analysis will be included in the multivariable analysis and later adjusted for confounding variables.

The confounding variables will be identified from the conceptual diagram as those variables that are thought to be associated with the candidate predicting variables and all causes 3 months post-discharge mortality. The association of every potential confounder with the all causes, 3 months post-discharge mortality will be established using a likelihood ratio test.

The survival rate will be displayed using the Kaplan-Meier curve, and a log-rank test will be used to compare the mortality rates between different diagnosis categories, age groups (5–9 years versus 10–14 years), and gender. Hazard ratios and their 95% confidence intervals will be reported with a significance level of 5%. The missing data will be assessed for any potential biases and excluded in the analysis.

## Ethical consideration

The study received ethical clearance from Muhimbili University of Health and Allied Sciences institutional review board with ethical clearance number 04-2023-1662.

## Discussion

The study is intended to investigate the burden of post-discharge mortality among children aged 5–14 years in Tanzania. It also aims to determine the predicting factors for mortality 3 months after discharge from the hospital among children aged 5–14 years. The clinical characteristics, demographics, and socioeconomic factors are the candidate variables for clinical prediction. The candidate variables for clinical prediction will all be analyzed to associate their impact with the primary outcome.

The primary outcome will be measured as incidence. The study is also intended to study several other secondary outcomes including the readmission rate, and the number of unplanned hospital/outpatient visits after discharge from the hospital with the predicting factors. The results from this study will bring awareness to clinicians about post-discharge mortality and its predicting factors in children aged 5–14 years. The results are also expected to form baseline data for subsequent studies that will establish a prediction model for post-discharge mortality in children aged 5–14 years.

The cohort design allows determining the effect of multiple predicting variables on three months post-discharge mortality. The study will use a large sample size from three different tertiary facilities that receive patients from all over Tanzania, hence the results can be generalized. The enrollment of the participants will be convenient to avoid bias. Research assistants will be medical officers who are not involved directly with the care of the patients to minimize bias. The follow-up period is chosen to be three months since a majority of the deaths related to previous admissions are reported to occur early in the months following discharge. The longer the follow up the more likely to get unrelated causes where results interpretation will be difficult.

The results are expected to reveal the burden the post-discharge mortality and its predictors. The results will determine which clinical characteristics can be directly associated with high post-discharge mortality and which factors can be avoided to reduce the post-discharge mortality. The results will also determine at which time post-discharge most deaths occur and what are the major causes. Referring to the recent study from Northern Tanzania that included

children aged 2–12 years, it is expected that chronic illnesses to be great predictors for post-discharge mortality. However, this study did not provide disaggregated data for children aged 5–14 years.

However, there are a few limitations to this study which are expected including the observational nature of the study makes it prone to miss some of the data due to lack of documentation. This could happen due to a lack of routine documentation of all patient clinical information most often due to a lack of necessary gadgets to measure some clinical parameters such as blood pressure.

It will also be difficult to establish the exact cause of death as a verbal autopsy will be conducted over the phone, and hence miss a chance to see some of the documents that could help in establishing the cause of death. Such documents include a death certificate or any hospital records and medication prescriptions that would help determine the possible cause of death. To overcome the limitation of over-the-phone verbal autopsy, for the deceased who died in the hospital the study is being conducted, and their documents will be collected from the hospital records.

## Acknowledgments

I would like to acknowledge the Muhimbili University of Health and Allied Sciences for providing me with access to HINARI which has enabled me to access various literature from several databases including PubMed. The university provides the log in credentials to all students for HINARI.

## Author Contributions

**Conceptualization:** Elton Roman Meleki.

**Supervision:** Stella Mongella, Francis Fredrick Furia.

**Writing – original draft:** Elton Roman Meleki.

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
