## [Decision Letter · Decision Letter 0]

18 Oct 2023

PONE-D-23-16416A STUDY PROTOCOL FOR PREDICTORS OF POST-DISCHARGE MORTALITY AMONG CHILDREN AGED 5-14 YEARS ADMITTED TO TERTIARY HOSPITALS IN TANZANIA: A PROSPECTIVE OBSERVATIONAL COHORT STUDY.PLOS ONE

Dear Dr. MELEKI,

Thank you for submitting your manuscript to PLOS ONE. After careful consideration, we feel that it has merit but does not fully meet PLOS ONE’s publication criteria as it currently stands. Therefore, we invite you to submit a revised version of the manuscript that addresses the points raised during the review process.

We look forward to receiving your revised manuscript.

Kind regards,

Deogratias Munube

Academic Editor

PLOS ONE

Journal Requirements:

Additional Editor Comments:

Dear Author,

Thank you for the submission. I have provided a few comments to improve your manuscript. Starting with the abstract. this is inconsistent with the whole body of the study protocol. The protocol needs a review of the English grammar. The methods described needs to be improved. For example, the method of recruitment is inconsistent. In one sentence, it is reported that the participants will be conveniently recruited and in another sentence consecutively recruited. The other need to be consistent. The justification for the study does include lack of studies in sub Saharan Africa. However, there are several studies referenced in East Africa. The authors need to improve the justification for the study. The list of abbreviations needs to be listed in alphabetical order. The author indicates that he did not receive any funds but later states that his hospital provided funds. This needs to be corrected. The author acknowledges the contribution of the local university for providing HINARI, this needs to be clarified. Did the university buy the HINARI account for the author. The author does indicate that this is his masters thesis project and does not describe if this is a requirement for his university to publish the protocol? The outcomes stated in the protocol is not consistent. The author needs to correct the impression that the initial primary outcome varies in the manuscript. The study sites need to be clearly described such that a reader understands why the three hospital sites need to be used. The process of recruitment and follow up needs to be clearly described. The telephone call script needs to be added to the protocol. The verbal autopsy guides needs to be included in the manuscript. For the confirmation of the possible cause of death at home or in the community, an document needs to be added to show the reader what process will be used to get that information from the parent or care giver. In the submission, it is indicated that there are no data tables submitted but in the manuscript there are two tables referenced. Are these dummy tables?

Reviewers' comments:

Reviewer's Responses to Questions

**Comments to the Author**

1. Does the manuscript provide a valid rationale for the proposed study, with clearly identified and justified research questions?

Reviewer #1: Yes

2. Is the protocol technically sound and planned in a manner that will lead to a meaningful outcome and allow testing the stated hypotheses?

Reviewer #1: Partly

3. Is the methodology feasible and described in sufficient detail to allow the work to be replicable?

Reviewer #1: No

4. Have the authors described where all data underlying the findings will be made available when the study is complete?

Reviewer #1: No

5. Is the manuscript presented in an intelligible fashion and written in standard English?

Reviewer #1: Yes

6. Review Comments to the Author

You may also provide optional suggestions and comments to authors that they might find helpful in planning their study.

Reviewer #1: The study protocol provides a clear description of the research design and objectives. The authors aim to investigate post-discharge mortality among children aged 5-14 years in Tanzania and identify predictors of mortality at 3 months after discharge. The protocol outlines the study setting, sample size calculation, inclusion and exclusion criteria, variables to be collected, data collection methods, data management procedures, and data analysis plan.

The strengths of the study protocol include its prospective observational cohort design, which allows for the examination of multiple predictors on post-discharge mortality. The inclusion of three different tertiary hospitals increases the generalizability of the findings. The enrollment of consecutive participants and the involvement of research assistants who are not directly involved in patient care help minimize bias. The use of structured questionnaires and a modified verbal autopsy tool enhances data collection.

However, there are a few areas where the protocol could be improved.

1. The protocol lacks a clear research question or hypothesis, which would provide a focused direction for the study. Though some clues about what the study seeks to achieve can be found in the concluding part of the background, no hypothesis has been proposed to direct the focus of the study.

2. While the candidate variables for predicting post-discharge mortality are listed, there is no explicit explanation of their rationale or theoretical basis. Providing this information would strengthen the study's justification for selecting these variables.

3. The authors have not provided details about the data collection approach. The protocol could benefit from a more detailed description of the data collection process, including how the structured questionnaires will be administered and the training provided to the research assistants. It would also be helpful to clarify how missing data will be handled and accounted for in the analysis.

4. In the data analysis section, the protocol states that univariable and multivariable Cox regression analyses will be used to determine factors associated with mortality. However, there is no mention of potential confounding variables or how they will be addressed in the analysis. It would be important to consider potential confounders and adjust for them in the multivariable analysis to obtain more accurate estimates of the predictors' effects. A protocol must necessarily show more details about the study, accounting for possible limitations are hoe the

5. The discussion section provides a brief overview of the study's aims and potential implications, This section could be expanded to include a more thorough interpretation of the results to expect. The limitations section highlights some potential limitations, such as missing data and the challenges of conducting verbal autopsies over the phone. However, it would be beneficial to discuss these limitations in more detail and address strategies to mitigate them.

Overall, the study protocol in its current form does not meet the requirements of a study protocol. It is not detailed and elaborate enough to allow for replication of the study. By addressing the aforementioned areas of improvement, the protocol can be further strengthened to ensure the study's rigor and validity.

7. PLOS authors have the option to publish the peer review history of their article (what does this mean?). If published, this will include your full peer review and any attached files.

Reviewer #1: No

---

## [Author Response · Author response to Decision Letter 0]

7 Nov 2023

1. The protocol lacks research question/hypothesis- 

Response: The research questions have been added to the protocol. The research question has been made clear in the protocol. The questions have been stated on page 4.

2. Candidate variables lack theoretical basis

Response: A figure of legend has been added to the protocol page 8

3. The author has not provided details about data collection, how the questionnaire will be administered and how research assistants will be trained?

response: This has been explained page 9

4. In data analysis section it has not been stated how missing data will be addressed or confounders will be handled

Response: This has been addressed on page 11

5. The discussion section should be expanded to explain the expected results, limitations and how to mitigate them. 

Response: This has been explained on page 13

6.Clarification about Hinari 

Response: Page 15

7.Why was it necessary for the research to be conducted in three institutions?

Response: It is addressed on page 5

8. There is inconsistencies in the methodology, in some sentences it is written participants will be recruited conviniently and in the other sentence written will be recruited consecutively

Response: It has been responded on page 6

9. Grammar is not correct

Response: Grammar has been corrected using Grammarly app

10. Abbreviations not in alphabetical order

Response: corrections made page 14

11. Sample size has not been well explained.

Response: Addressed well on page 6

12. There was controversy on the statement about funding.

This has been addressed on page 14 and also on the cover letter corrections have been made.

---

## [Editor Report · Decision Letter 1]

7 May 2024

A STUDY PROTOCOL FOR PREDICTORS OF POST-DISCHARGE MORTALITY AMONG CHILDREN AGED 5-14 YEARS ADMITTED TO TERTIARY HOSPITALS IN TANZANIA: A PROSPECTIVE OBSERVATIONAL COHORT STUDY.

PONE-D-23-16416R1

Dear Dr. ELTON Roman Meleki

We’re pleased to inform you that your manuscript has been judged scientifically suitable for publication and will be formally accepted for publication once it meets all outstanding technical requirements.

Kind regards,

Deogratias Munube

Academic Editor

PLOS ONE

---

## [Editor Report · Acceptance letter]

10 May 2024

PONE-D-23-16416R1 

PLOS ONE

Dear Dr. Meleki, 

I'm pleased to inform you that your manuscript has been deemed suitable for publication in PLOS ONE. Congratulations! Your manuscript is now being handed over to our production team.

Kind regards, 

on behalf of

Dr. Deogratias Munube 

Academic Editor

PLOS ONE